# Water Jet Technology: Experimental Verification of the Input Factors Variation Influence on the Generated Vibration Levels and Frequency Spectra

**DOI:** 10.3390/ma14154281

**Published:** 2021-07-31

**Authors:** Stefania Olejarova, Tibor Krenicky

**Affiliations:** Department of Technical Systems Design and Monitoring, Faculty of Manufacturing Technologies, Technical University of Kosice, Sturova 31, 080 01 Presov, Slovakia; stefania.olejarova@tuke.sk

**Keywords:** abrasive water jet, vibrations, amplitude of vibration acceleration, frequency spectrum, type of abrasive, feed rate, mass flow of abrasive

## Abstract

Vibration measurement belongs amongst the most important activities for the correct and stable operation of machines in terms of quality, economy, and safety. Monitoring the condition of the machine provides the key data for the detection of machine parts damage, thus preventing unexpected failure states and production drop-outs. Therefore, special attention is paid in the paper to the measurement and evaluation of basic parameters of vibration, particularly the frequency of vibrations and the amplitude of vibration acceleration. Experimental measurements were performed while machining the material HARDOX 500/10 on a production system using the abrasive water jet technology (AWJ) with the varied abrasive mass flow and the feed rate. By evaluating the results of measurements, recommendations of suitable and inappropriate combinations of operating parameters were formulated, widening current knowledge in the field of the abrasive parameters and the speed of movement of the technological head influence the amplitudes of vibration acceleration in the operation of production systems with AWJ technology.

## 1. Introduction

Advanced technologies, also known as unconventional ones, are currently being implemented in order to increase the possibilities of machining difficult-to-machine and special materials. These technologies work on different principles than classical methods. Unconventional technologies use the electro-thermal principle, electrochemical, chemical, and mechanical principle for material removal. Technologies that use the mechanical principle also include high-pressure water jet machining technology, which provides precise, computer-controlled cold machining of materials, without mechanical deformation and with little waste. There is little waste here and mainly very effective usage of the material due to cold cutting with very narrow kerf resulting in a very small amount of scrap material. Secondarily, waste water and garnet usually do not contain harmful substances and thus can be recycled (used elsewhere).

The principle of water jet machining technology can be explained in terms of the material removal by mechanically impinging a liquid, usually containing solid particles for the efficiency improvement, on a workpiece [1,2]. With the usage of appropriate technological tools, the liquid has to be concentrated within a narrow jet flowing from the nozzle under sufficient pressure and acting on the material. The jet is gradually deflected against the cutting direction with the loss of its kinetic energy, which was consumed during the process of interaction with the machined material [3,4]. For cutting using a clean water jet, mostly pure water after chemical and mechanical treatment, without added mechanical particles, is used. The properties of liquid at high pressures (typically around 400 MPa) are used to form a highly effective cutting tool [5,6]. The abrasive water jet is one of the tools with an effect similar to grinding and the decisive mechanism for removing the material being machined is similar to that of the mentioned machining method. Cutting efficiency is enhanced by abrasive grains randomly distributed in the beam [7,8]. Most of the water jet cutting machines achieve high pressures using a multiplier. The principle of generating high pressures using a multiplier is based on the difference of the surfaces of two pistons firmly connected to each other [9,10]. Flexibility and cold cutting are some of the most advantageous properties of abrasive water jets, as well as the fact that this is an effective tool for cutting various materials including extremely hard or soft materials and composites and sandwich materials that are difficult to machine using traditional technologies [11,12,13,14,15].

Modal analysis along with monitoring of process data are aimed at studying the frequencies in which the vibration amplitude is significantly increased for certain induced frequencies, even when the periodic excitation forces are small [16,17]. In addition, since these and other processes affect the machining results, the impact of the changed conditions on the machining process needs to be further investigated in order to improve. The article is focused on the investigation of the influence of technological parameters on the magnitude of vibrations and their influence on the machine itself during the change of abrasiveness. Ongoing research and evaluation is aimed at improving machining results—meaning the quality of the material being machined. Therefore, the article is aimed at contributing to this effort, presenting the results from experimental observations of the influence of the mentioned factors on the magnitude of vibrations of the technological head.

The simplest method in order to increase the cutting performance in a waterjet abrasive machining program is to accelerate the process by increasing its parameters, primarily the pressure and cutting speed. The increase in these parameters, however, results in higher dynamic loads acting on the technological device. As a result, undesirable vibrations occur that reduce machining accuracy and cutting quality along with increasing the machine parts wear. Therefore, the dynamic characteristics (modal parameters) may be changed by the means of alteration of a construction design of a technological device [18,19]. The current state of knowledge on the process quality assessment based on vibration analysis is based primarily on vibration measurement and analysis directly during the process operation [20,21]. The model proposed by Monno and Ravasio [20] is based on assessment of the striation formation that depends mainly on the jet instability caused by vibrations during the cutting process. Vibration signals have been measured whilst varying the cutting conditions. A model has been developed by evaluation of the mean spacing of the striations as a function of the period and the amplitude of the jet vibration. The results of the research reported by Krenicky and Rimar [21] summarize the evaluation of vibrations generated during the AWJ machining. On that basis, the means to minimize the influence of cutting-head vibrations on the machined work-piece surface quality are discussed. The novelty of the present manuscript is based on a unique combination of the thick hard material and operational parameters, and a combination of the evaluation methods. The authors believe that this combination can be useful from some points of view—research as well as practice. The frequency and amplitude of vibration during the cutting process depends to a great extent on the workpiece thickness [22]. From the other parameters, the vibration pattern is strongly influenced by the diameter of the cutting jet—with the increase of the jet diameter, the spectral density of the power is transferred to the lower frequencies [23]. The frequency ranges analyzed in the studies depend on the process parameters [24]. An important multi-parametric description of the cutting process has been presented by the Ostrava group [16,17,18,19,25]. The group of TU Kosice researchers is involved in systematic studies of the operational states of manufacturing processes using progressive technologies and the influence of the process parameters on the operation and quality of the machining [26,27,28]. The topic of the study [26] is closely related to the present article topic using the same equipment, but authors are now bringing a unique combination of the materials and parameters in order to enhance the database of the results.

Although there is a constantly growing set of developed solutions to the problem, including methodologies and evaluations of experiments valid for specific measurement conditions, the current solutions still do not cover several variations. Abrasive water-jet cutting is a multiparametric process where the quality of the output characteristics relies on the inputs. This has been proven by various experiments and theoretical analyses, and the authors also personally know that topic. However, there are simultaneously numerous attempts of creating the models that characterize the cut area quality under specific conditions with regard to material type and thickness; technology head traverse speed and other parameters; the abrasive mass flow rate, grain size and many others. Our work is aimed to complement data for models present in some other research works as some readers might appreciate such kind of information. In our opinion, the study of parameters in the presented combination and range are unique. Thus, the main aim of the presented research is to contribute to the clarification of the issues of the relationship between the input technological factors and the resulting vibration parameters.

## 2. Materials and Methods

The experimental measurements were performed at a research experimental workplace in the Laboratory of Liquid Jet, Institute of Physics, Faculty of Mining and Geology, the Technical University of Ostrava. The experiments were performed on an XY CNC WJ1020-1Z-EKO workbench (Figure 1) for planar applications using the abrasive water jet cutting technology. The PTV 19/60 HSQ 5x type multiplier was used to generate water pressure with a flow rate of up to 1.9 L/min. All measurements were performed from the same starting position with coordinates X = 320 mm and Y = 370 mm. Twenty-second samples were selected from the time course, and the frequency spectra generated by the Fourier transform were evaluated in the range up to 10 kHz.

A PASER IIITM type cutting head with a water nozzle diameter of 0.25 mm and a focusing tube diameter of 1.02 mm was used to cut the special steel material K13—HARDOX 500/10 with thickness of 10 mm, manufactured by SSAB, Borlänge, Sweden. The working medium pressure of 380 MPa and the grain size of the selected MESH 80 abrasives during the measurement were constant [29]. We decided that, because this part of the work was an additional part of the series of similar research related to one thematic area, the authors aimed this solution to be supplementation of the research reported e.g., in the dissertation thesis [29]. Thus, a more detailed description of the experimental design can be found in the previous works [26,29]. The size of the abrasive particles significantly affects the AWJ mechanism in terms of their inlet into the water jet and the kinetic energy required to cut the material. Circularity and roundness express the shape of abrasive particles. Circularity is the largest measured distance of the true circle from the enveloping circle. The roundness is defined in a similar way, with the only difference that it is the distance of the real ball from the enveloping ball. High roundness and low circularity are important prerequisites for the best possible shape of abrasive particles. The material properties of both AG and UG abrasives used are similar, as it is an abrasive material—garnet with hardness according to KNOOPA 1350, a relative hardness of 1, and a density of 3.8 g/cm. The circularity reaches a value of about 0.48 and the roundness of about 0.78 [30]. Alluvial Garnet is a mineral sand, and how you get it in the bag is how it was in the ground (all the impurities and other minerals removed obviously). Australian Garnet is in this category—it is low dust, is pink in color, and has a ‘sub-angular’ shape that is perfect for blasting. Ukrainian Garnet is a rock type mineral, and has to be crushed to be a suitable size for use when abrasive blasting. Also known as ‘hard rock garnet’, the crushing process puts minute facture lines in all the garnet particles, which means it is very dusty when blasting and does not recycle very well.

The distance of the nozzle above the workpiece was also the same for all measurements, namely 2 mm. As changing input values, the type of abrasive was chosen, namely Australian garnet (AG) and Ukrainian garnet (UG), feed rate 50 and 100 mm/min; abrasive mass flow of 400, 500, and 600 g/min in particular experiments [26,30]. The plan was created to complement the solutions of previous work, using the parameters variation of available equipment and cutting material that is interesting both for research and practice. Finally, it is a planned experiment but not a full factorial one, as not every input factor was used at each level for each measurement. Figure 2 shows samples of MESH 80 grain size of abrasives used under a microscope using a magnification of 50 times [29].

The piezoelectric accelerometer type Bruel & Kjaer 4514-B, Nærum, Danemarca was used to measure the magnitude of vibrations during the experiments. The basic technical parameters of the sensor are listed in Table 1.

Data acquisition was performed using a simultaneously sampled 4-channel AI module NI-9233. The main technical parameters of the DAQ module are listed in Table 2.

The LabVIEW SignalExpress software system by National Instruments (Austin, TX, USA) is a complex tool for collecting, evaluating, and archiving data suitable for the field of machine monitoring, also providing tools for the vibration diagnostics. It contains scalable functions and tools necessary for signal analysis in the time and frequency domains. From the measurement record of a stabile signal, a representative part of 20 s was selected from a total time record of 120 s, and a frequency spectrum in the range of 0 to 10 kHz was evaluated using a Fourier transform.

Before starting the measurement, it was necessary to determine the location of the piezoelectric sensor. Whereas fastening by screwing to the technological head of the water jet device is not possible, the sensor was fixed with a thin layer of inelastic glue (Figure 3) so that its axis coincides with the axis of vibration in the direction of the abrasive water jet.

The evaluation of the measured data from the performed vibration measurements consisted firstly of the recording of the measured records, the raw data from the measuring instrument to the computer. Then, data were processed using the SignalExpress ver. 7.0.0 software (National Instruments, Austin, TX, USA), particularly FFT analysis tool. The results are recorded in the form of frequency spectra of vibration acceleration and graphical dependence for predetermined input values.

## 3. Results

### 3.1. Evaluation of Measurement While Cutting Using AG

The graph in Figure 4 shows the dependence of the vibration acceleration on the frequency up to 10 kHz for the scanning point of the technological head when machining with predetermined mass flows of abrasive AG 400, 500, and 600 g/min and the feed rate of 50 mm/min.

An evaluation of the frequency spectrum when cutting material using a mass flow of 400 g/min shows that significant vibration amplitudes can be observed in the following frequency ranges from 1.4 to 1.6 kHz; from 2.3 to 3.5 kHz; from 4.3 to 4.6 kHz; and from 5.2 to 6.5 kHz. The dominant value of the vibration amplitude is around 0.322 mg at a frequency of 5.4 kHz. At a mass flow of 500 g/min, the significant vibration levels can be observed in two frequency ranges from 2.3 to 3.5 kHz and from 5.0 to 5.7 kHz, where the peak value of the vibration amplitude reached 0.425 mg at a frequency of 5.4 kHz. In comparison with the aforementioned mass flows, it can be seen that, using a flow of 600 g/min, the level of the vibrations is comparable to that of a flow of 400 g/min with the peak value of vibration amplitude 0.331 mg at a frequency of 5.4 kHz.

Figure 5 shows the frequency dependence of the vibration acceleration in the range 0–10 kHz for the measurement node on the technological head at cutting with predetermined mass flows of abrasive AG 400, 500, and 600 g/min and the feed rate of 100 mm/min.

Evaluation of the frequency spectrum measured at cutting material using the AG abrasive mass flow of 400 g/min, it can be seen that more significant vibration amplitudes were achieved in three frequency ranges from 1.5 kHz; from 2.0 to 3.5 kHz; and from 4.0 to 6.5 kHz. The dominant value of the vibration amplitude is around 0.326 mg at a frequency of 2.7 kHz. When cutting the material at all aforementioned experimental mass flow values, dominant values of the amplitude of the vibration acceleration were achieved at the same frequency around 2.7 kHz. At a mass flow of 500 g/min, more significant vibration values were achieved in two frequency ranges from 2.0 to 3.5 kHz and from 5.0 to 6.3 kHz where the peak value of the vibration amplitude reached 0.314 mg at a frequency of 2.8 kHz. At a flow rate of 600 g/min, a dominant vibration value was achieved approximately the same as when cutting by flows of 400 and 500 g/min, namely 0.347 mg at a frequency of 2.7 kHz.

The envelope method was used to evaluate the levels of the measured values. The envelopes are designed for a better overview in the frequency range 2000–8000 Hz only because, from the previous waveforms in Figure 4 and Figure 5, it can be seen that the dominant values of the vibration acceleration amplitude are generated in this range. Figure 6 shows a comparison of the measured values after the creation of graphical dependences separately for the selected mass flows AG and together for the feed rate.

After evaluating the graphical dependences from the envelopes, it is clear that the vibrations at both feed rates used reached approximately the same value. The difference between them was minimal, and, in this case, it can be neglected. Therefore, when machining using AG, it is equally advantageous to use both feed rates of the process head as required. During machining, a mass flow of 400 g/min, and a feed rate of 100 mm/min, the value of this amplitude is 1.01 times higher than when divided by a speed of 50 mm/min. At a flow rate of 500 g/min, the value of the amplitude of the vibration acceleration at a speed of 100 mm/min is 1.36 times lower than at a speed of 50 mm/min. Finally, at a flow rate of 600 g/min at a feed rate of 100 mm/min, the amplitude value is 1.05 times higher compared to the value obtained at a speed of 50 mm/min.

### 3.2. Evaluation of Measurement While Cutting Using UG

Figure 7 presents the frequency dependence of the vibration acceleration in the range up to 10 kHz for the measured node on the technological head when cutting using the predetermined mass flows of abrasive UG 400, 500, and 600 g/min, and a feed rate of 50 mm/min.

Evaluation of the frequency spectrum measured while cutting the material using the abrasive mass flow of 400 g/min reveals that significant vibration amplitudes were generated in two frequency ranges from 5.0 to 5.7 kHz and from 7.0 to 10.0 kHz. The dominant values of the vibration amplitude were increasing up to the value of 0.792 mg at a frequency of 10.0 kHz. When cutting the material at all selected mass flows, dominant values of the amplitude of the vibration acceleration were observed at the same, highest measured frequency, namely 10 kHz. At a mass flow of 500 g/min, more significant vibration values were also achieved in two of the same frequency ranges as at a flow of 400 g/min, whilst the highest value of the vibration amplitude reached the value of 0.915 mg. At a flow rate of 600 g/min, the highest vibration value of 0.959 mg was reached.

The graph in Figure 8 shows the frequency dependence of the vibration acceleration in the band up to 10 kHz for the measurement node on the technological head when cutting using predetermined mass flows of abrasive UG 400, 500, and 600 g/min and the feed rate of 100 mm/min.

An examination of the frequency spectrum when cutting a material with a mass flow of 400 g/min shows that significant vibration amplitudes were generated within one frequency range from 4.8 to 9.0 kHz. The peak value of the vibration amplitude is around 0.601 mg at a frequency of 7.3 kHz. At a mass flow of 500 g/min, more significant vibration values were also observed in one and the same frequency range as when using a mass flow of 400 g/min where the peak value of the vibration amplitude reached the value of 0.534 mg at a frequency of 6.9 kHz. Compared to the previous mass flow rates used for the cutting, it can be concluded that, at a flow rate of 600 g/min, the vibration value is significantly lower, and its peak value of the vibration amplitude is around 0.618 mg at a frequency of 6.3 kHz.

Similar to the previous case, the envelope method was used to evaluate the measured spectra. Envelopes are designed for better clarity only in the frequency range 4800–10,000 Hz, since, from the previous spectra in Figure 6 and Figure 7, it can be seen that the dominant values of the amplitude of the vibration acceleration are recorded in this range. The comparison of measured values consists in the creation of graphical dependences which are presented in Figure 9 separately for the selected UG mass flows and together for the feed rate.

However, the assumption that an increased level of vibration will be observed when cutting with UG abrasive at a feed rate of 100 mm/min was not confirmed after evaluating the graphical dependences of the envelopes. Higher values were obtained when cutting at a speed of 50 mm/min, where the dominant value of the amplitudes of the vibration acceleration occurs in all measurements at the highest measured frequency (10 kHz). When cutting using a mass flow of 400 g/min and a feed rate of 100 mm/min, the value of this amplitude is 1.32 times lower than when divided by a speed of 50 mm/min. At a flow rate of 500 g/min, the value of the amplitude of the vibration acceleration at a speed of 100 mm/min is 1.72 times lower than at a speed of 50 mm/min. Finally, at a flow rate of 600 g/min at a feed rate of 100 mm/min, the amplitude value is 1.55 times lower compared to the value obtained at a speed of 50 mm/min.

## 4. Analysis and Discussion

### 4.1. Analysis of Signal Parameters

The main vibration parameters examined are Root Mean Square (RMS) and Peak to Peak of the instantaneous values in a certain time duration. It relates to the power of the wave. The RMS value of velocity is one of the important factors for machinery status diagnosis. Peak to Peak is the difference between the maximum positive and the maximum negative amplitudes of a waveform. It is the total distance passed by a vibrating body from one extreme to the other. RMS vibration is calculated by measuring the peak amplitude and multiplying by 0.707 to obtain the RMS value. Finally, divide by 0.707 to get the Peak to Peak. These parameters were used to analyze the comparison of the vibration signal for selected mass flows of the abrasive AG, UG, and the feed rate of the technological head, which received more attention. During signal processing, different values of these monitored parameters were observed. Differences point to the influence of unidentified factors related to the tool—the technological head. Verification of the influence of the factors forming the abrasive water jet cutting tool on the measured values should be the subject of further research, as it is a technology requiring a multi-parametric approach.

### 4.2. RMS Value

The graph in Figure 10 presents the RMS values as a function of the mass flows of the abrasive AG when cutting with the feed rates of 50 and 100 mm/min. The curve corresponding to a feed rate of 50 mm/min shows a significantly higher value at an abrasive mass flow of 500 g/min. Approximately the same RMS values were recorded at a speed of 50 mm/min and also 100 mm/min corresponding to the abrasive mass flows of 400 g/min and 600 g/min.

The following graph of Figure 11 presents the RMS values at the same factor settings as in the previous case. The difference is based on replacing the AG abrasive with UG. Significantly higher peaks were generated at a rate of 50 mm/min. From that curve, it can be concluded that, with increasing mass flow of the abrasive, an increasing trend of values is observed. The smoother course was achieved when cutting with the feed rate of the technological head of 100 mm/min.

### 4.3. Peak to Peak Value

Another monitored vibration parameter was the span of values—Peak to Peak. The evaluation of these values as a function of the mass flows of the abrasives AG and UG is shown in Figure 12 and Figure 13. The course of the curves is more organized compared to the previously investigated RMS parameter. From the course of the curves, it can be stated that, with increasing mass flow of the abrasive, an increasing trend of values is also recorded. The lowest Peak to Peak value was recorded for both types of abrasive AG and UG used at an abrasive mass flow factor of 400 g/min and a feed rate of 100 m/min. The highest Peak to Peak value was observed using a UG abrasive and a feed rate of 50 mm/min.

The deviations that can be seen in the graphs presented in Figure 10, Figure 11, Figure 12 and Figure 13 when machining with AG and UG abrasives are presumably caused by the circumstances that occurred during the cutting of Hardox 500 material:the grid has been cut;a cut was led outside the basic ribbing in the attenuating tank;the jet was led so that cut also reached through the ribs of the tank;a drop in water level occurred;the multiplier has skipped.

### 4.4. Discussion on the Possible Continuation of Research in This Area

From the analysis of experimentally obtained results of the influence of technological parameters and from the point of view of optimization of the abrasive water jet cutting process on the XY CNC WJ1020-1Z-EKO workbench, it is necessary to state that the most unfavourable combination of monitored parameters clearly confirmed the combination when cutting with UG abrasive at all mass flows examined. In this combination, the highest values of vibration acceleration amplitudes were generated, which in long-term operation may cause an increase in the duration of fault downtimes and consequently a decrease in the service life of selected functionally important parts of the water jet production system. It also negatively affects the quality of the machined surface of the material, which leads to an increase in the roughness of the cut and also to an increase in the laboriousness of finishing the products [26]. Last but not least, it also increases the time and environmental burden of the production process. Under such operating conditions, the combination of unsuitable factors has the effect of reducing the economic efficiency of the operation in connection with increasing its operating costs and reducing the quality of production.

On the other hand, within the investigated set of input factors, it is recommended to machine the material using AG abrasive, the speed of the technological head 100 mm/min for all investigated mass flows. With this combination of factors, approximately the same values of vibration acceleration amplitudes have been achieved, which, particularly in long-term operation, have an adverse effect on the reliability, service life, operating costs, and operational safety of the production system with water jet technology.

From the research results, it is concluded that AG is more suitable for the use in machining, where lower levels of vibrations have been achieved, but, on the other side, the quality of the machined material may not be sufficient. Therefore, it is necessary to take into consideration for what applications it is more advantageous to use AG, even if the machine operation is not affected from vibrations at a high level, but we do not achieve high surface quality, or to use another type of abrasive for which the amplitudes of vibration are not too large to cause rapid wear to the machine parts. This may be the subject of further research.

## 5. Conclusions

Due to its characteristics, the abrasive water jet cutting technology is one of the sought-after options for cutting materials. Despite its advantages, like any other technology, it has also some weaknesses. The development of innovations to eliminate some shortcomings is constantly the subject of research. 

Significantly increased amplitudes in the low frequency portions of the spectrum were observed in frequency analysis using the Australian garnet (AG) abrasive. When the type of abrasive was changed to Ukrainian garnet (UG), the increased amplitudes shifted to the high-frequency parts of the spectrum.

The knowledge gained from the analysis of experimental data can be used in further experiments and also in optimizing the operation of production facilities using abrasive water jet technology for material machining. It will be necessary to focus on the selection of the most suitable types of abrasives and the most suitable types of abrasive mass flows, which will guarantee the lowest levels of vibrations of the technological head and thus more economically efficient cutting of materials. Since the speed of movement of the technological head also has a certain influence on the vibration of the technological head, another possible direction of research is to determine the optimal speed when dividing materials by this technology under particular conditions. The correct qualification and vibration analysis of selected parts of production systems with AWJ or in laboratory or operating conditions serves as a basis for design solutions aimed at suppressing the occurrence and propagation of undesirable vibrations of individual parts of the system, especially the technological head.

To conclude, this paper is aimed at contributing to the clarification of the issues of the relationship between the input technological factors and the resulting vibration parameters investigated on the technological head. 

## Figures and Tables

**Figure 1 materials-14-04281-f001:**
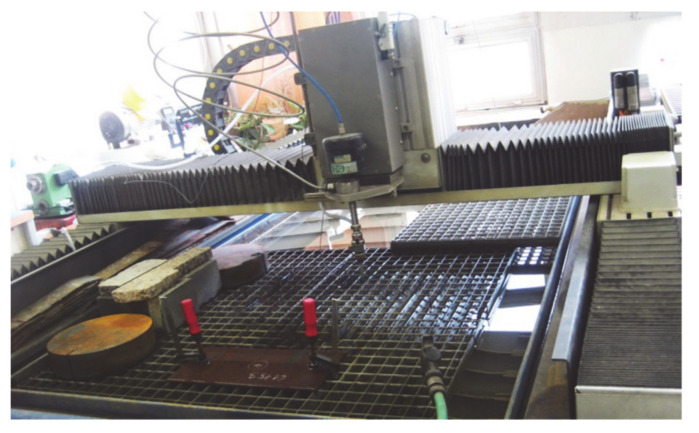
Experimental system with water jet technology in the laboratory of liquid jet.

**Figure 2 materials-14-04281-f002:**
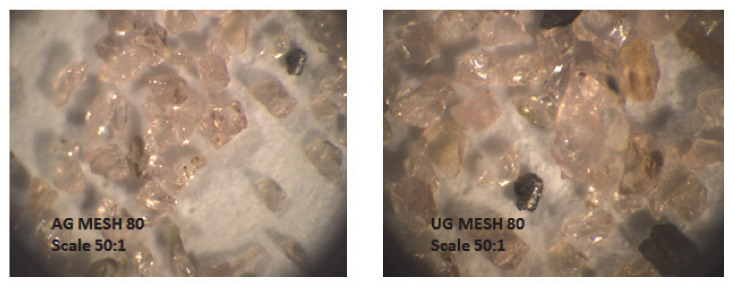
AG and UG samples with MESH 80 grain size.

**Figure 3 materials-14-04281-f003:**
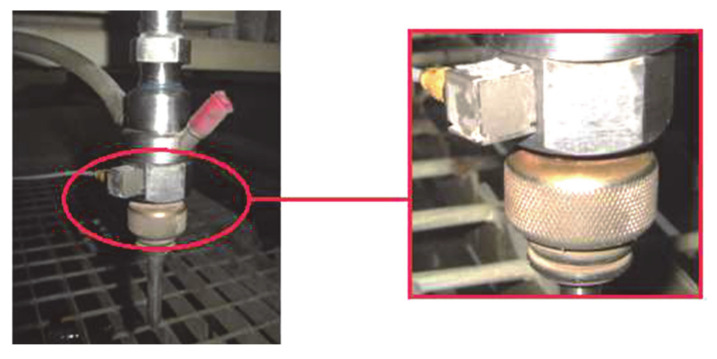
Detailed image of the mounting of a miniature vibration acceleration sensor on the technological head.

**Figure 4 materials-14-04281-f004:**
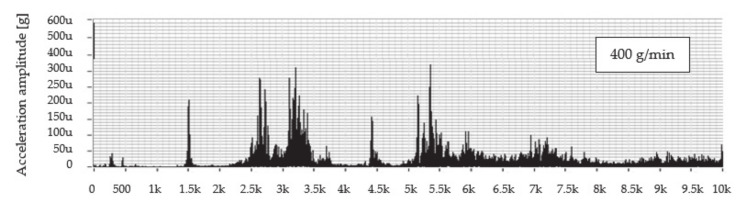
Frequency spectra of vibration acceleration amplitude versus frequency for abrasive mass flows at a feed rate of 50 mm/min for abrasive AG.

**Figure 5 materials-14-04281-f005:**
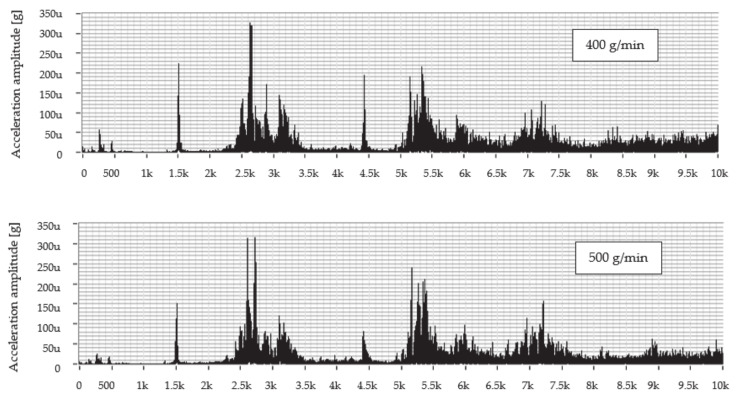
Frequency spectra of vibration acceleration amplitude versus frequency for abrasive mass flows at a feed rate of 100 mm/min for abrasive AG.

**Figure 6 materials-14-04281-f006:**
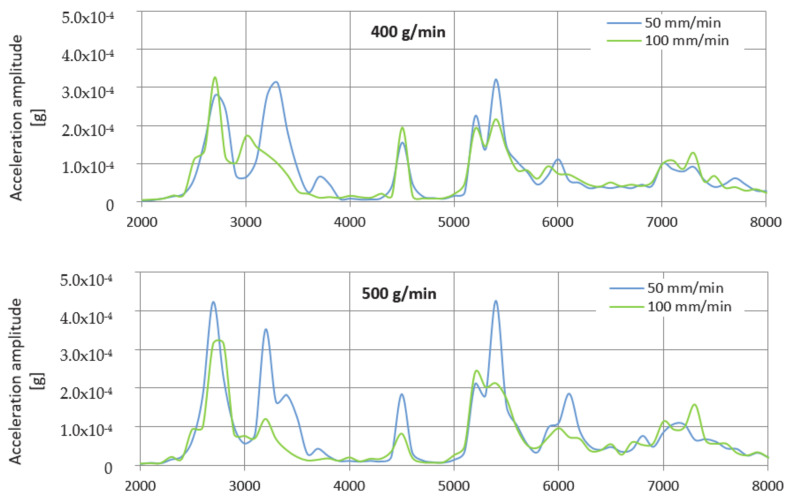
Graphs comparing envelopes of frequency spectra when cutting with an AG.

**Figure 7 materials-14-04281-f007:**
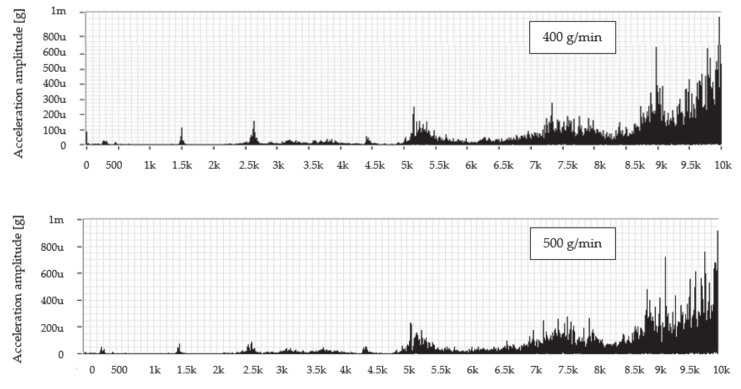
Frequency spectra of vibration acceleration amplitude versus frequency for abrasive mass flows at a feed rate of 50 mm/min for abrasive UG.

**Figure 8 materials-14-04281-f008:**
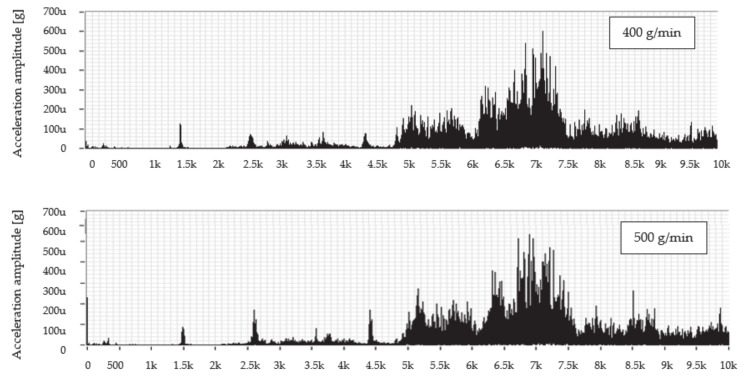
Frequency spectra of vibration acceleration amplitude versus frequency for abrasive mass flows at a feed rate of 100 mm/min for abrasive UG.

**Figure 9 materials-14-04281-f009:**
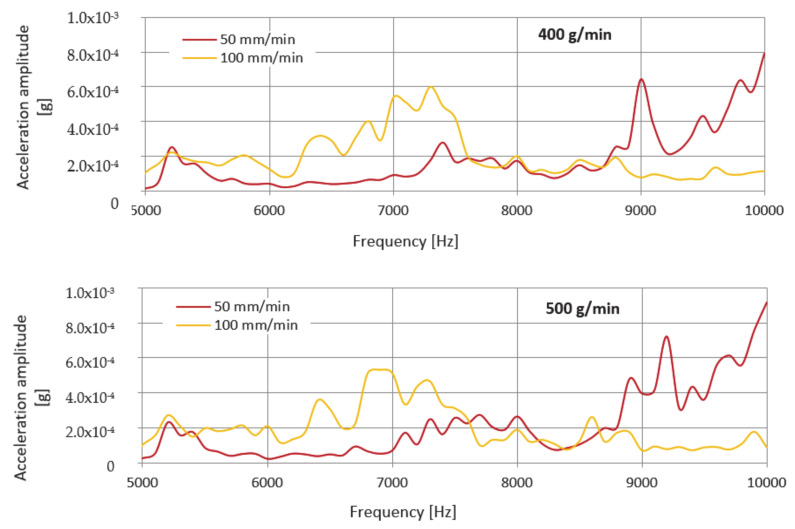
Graphs comparing envelopes of frequency spectra when cutting with an UG.

**Figure 10 materials-14-04281-f010:**
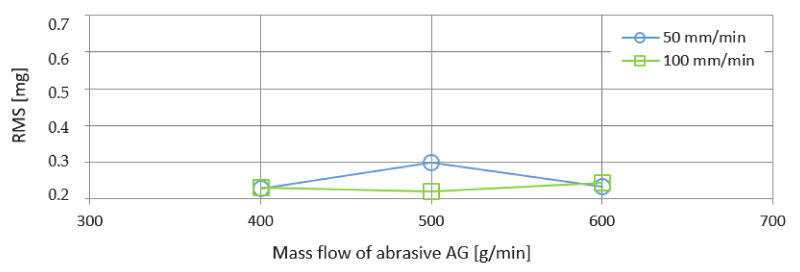
Graphical dependence of RMS on mass flows of abrasive AG.

**Figure 11 materials-14-04281-f011:**
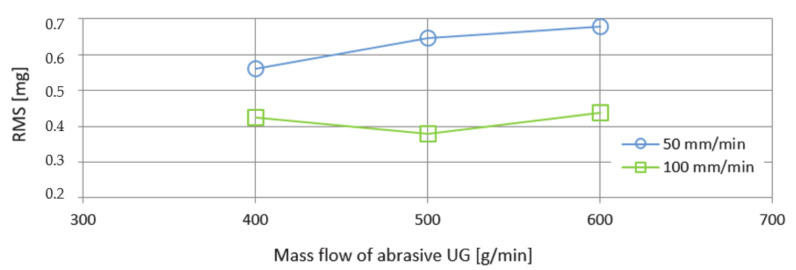
Graphical dependence of RMS on mass flows of abrasive UG.

**Figure 12 materials-14-04281-f012:**
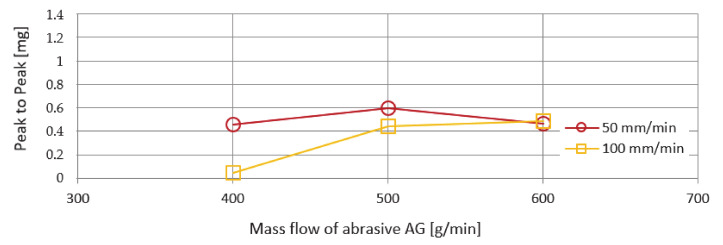
Graphical dependence of Peak-to-Peak on mass flows of abrasive AG.

**Figure 13 materials-14-04281-f013:**
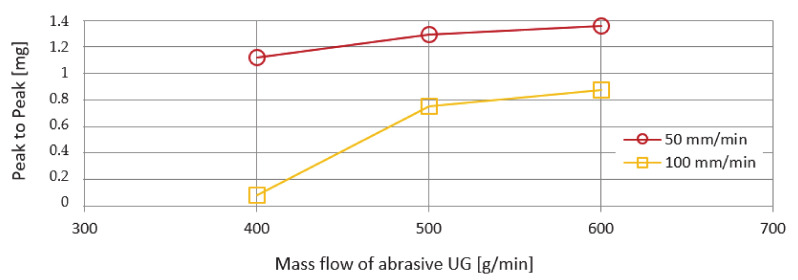
Graphical dependence of Peak-to-Peak on mass flows of abrasive UG.

**Table 1 materials-14-04281-t001:** Selected technical parameters of the Bruel & Kjaer 4514-B accelerometer.

Measuring range (±pk) [m·s^−2^]	4900
Nominal voltage sensitivity (at 160 Hz) [mV/g]	10
Frequency range [Hz]	1–10 k
Start-up time [s]	1

**Table 2 materials-14-04281-t002:** Selected technical parameters of the NI-9233 DAQ module.

ADC resolution [bit]	24
Input range [V]	±5
Dynamic range [dB]	102
Max. sampling rate [Sa/s]	50 k
Time base clock frequency [Hz]	12.8 M

## Data Availability

The data that support the findings of this study are available from the corresponding author upon reasonable request.

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
