# Peer review of "Water Jet Technology: Experimental Verification of the Input Factors Variation Influence on the Generated Vibration Levels and Frequency Spectra"

_materials, 2021, doi:10.3390/ma14154281_

Round 1

Reviewer 1 Report

The goal is to contribute to the clarification of the relationship between the input technological factors and the resulting vibration parameters.
Selected as variable input values:
Abrasive type: Australian garnet (AG) and Ukrainian garnet (UG),
Feeding speed 50 and 100 mm / min;
The mass flow rate of the abrasive is 400, 500 and 600 g / min in specific experiments.

1. Literary review. Consists primarily of contemporary sources. At the same time, the review is very superficial. The well-known concepts of the Water jet technology and modal analysis are presented.

2. Materials and methods. Research methods are generally accepted and adequately selected.
As a side note. It is necessary to indicate the frequency of recording signals.

3. Results. The frequency characteristics of vibration signals are given.

4. Discussion.
The discussion is more like a statement of observed facts than a discussion of the results.
There is absolutely no explanation of the reasons for the change in RMS and Peak-to-peak with changes in the studied technological parameters. Changes are clearly visible on charts 9-12, but the authors do not provide an explanation of the reasons for these changes. The analysis of the reasons for the change in the RMS behavior when changing the abrasive from AG to UG is not provided (as can be seen from graphs 9 and 10).
5. Conclusions.
The conclusions are not specific and require substantial revision.
Authors in the conclusions provide more discussions of the results than in section 4. These discussions should be removed and only specific results / recommendations left.

General conclusion based on the review.
I believe that the authors should significantly revise their manuscript. Move the reasoning from the conclusions to section 4 and combine with the data presented there. It is also worthwhile to slightly expand the analysis of the results to better understand the reasons for these changes when changing the type of abrasive and processing modes.
Taking into account the subject matter of the journal, images of the abrasive (AG and UG), peculiarities of their composition, morphology, etc. can be given. Based on this, try to analyze the results and try to identify the reasons for the change in the RMS behavior when changing the abrasive from AG to UG (as can be seen from graphs 9 and 10).

Author Response

Dear Reviewer,

we are very grateful for your comments.

Regarding the comments from your review, we tried to respond to them as best we could and added recommended improvements and explanations into the manuscript. For further details, please see the attachment.

Best regards,                                                                                                         Authors

Reviewer 2 Report

This is a very good paper.

However there is a notice: The denominations of Figure 6 and Figure 7 are the same. "Frequency spectra of vibration acceleration amplitude versus frequency for abrasive mass flows at a feed rate of 50 mm / min for abrasive UG".
Seeing the structure of the paper the last two letters of Figure 6 should be "AG".

Author Response

(The authors gave the same response as above.)

Reviewer 3 Report

See the attachment for comments.

Author Response

Dear Reviewer,

we are very grateful for your comments.

Regarding the questions and comments from your review, we tried to respond to them as best we could and added recommended improvements and explanations into the manuscript. For further details, please see the attachment.

Best regards,                                                                                                         Authors

Round 2

Reviewer 1 Report

The remarks have been eliminated. Article significantly revised and can be recommended for publication.

Author Response

Dear reviewer,

thank you so much again for your comments.

Best regards,

authors

Reviewer 3 Report

See the attachment for comments.

Author Response

Dear reviewer,

thank you so much for your valuable comments and suggestions aimed at significant improvement of our manuscript.

For detailed responses, please see the attachment.

Best regards, 

authors
